# Controlled Swelling of Monolithic Films as a Facile Approach to the Synthesis of UHMWPE Membranes

**DOI:** 10.3390/membranes13040422

**Published:** 2023-04-09

**Authors:** Konstantin Pochivalov, Andrey Basko, Tatyana Lebedeva, Mikhail Yurov, Alexey Yushkin, Alexey Volkov, Sergei Bronnikov

**Affiliations:** 1G.A. Krestov Institute of Solution Chemistry of the Russian Academy of Sciences, 1 ul. Akademicheskaya, 153045 Ivanovo, Russia; avb@isc-ras.ru (A.B.);; 2A.V. Topchiev Institute of Petrochemical Synthesis of the Russian Academy of Sciences, 29 Leninsky Prospect, 119991 Moscow, Russia; 3Biological and Environmental Science, and Engineering Division (BESE), Advanced Membranes and Porous Materials Center (AMPM), King Abdullah University of Science and Technology, Thuwal 23955, Saudi Arabia; 4Institute of Macromolecular Compounds of the Russian Academy of Sciences, 31 Bolshoy pr., 199004 St. Petersburg, Russia

**Keywords:** ultra-high molecular weight polyethylene, ultrafiltration membrane, thermally induced phase separation, swelling, semicrystalline polymer, phase diagram

## Abstract

A new method of fabricating porous membranes based on ultra-high molecular weight polyethylene (UHMWPE) by controlled swelling of the dense film was proposed and successfully utilized. The principle of this method is based on the swelling of non-porous UHMWPE film in organic solvent at elevated temperatures, followed by its cooling and further extraction of organic solvent, resulting in the formation of the porous membrane. In this work, we used commercial UHMWPE film (thickness 155 μm) and o-xylene as a solvent. Either homogeneous mixtures of the polymer melt and solvent or thermoreversible gels with crystallites acting as crosslinks of the inter-macromolecular network (swollen semicrystalline polymer) can be obtained at different soaking times. It was shown that the porous structure and filtration performance of the membranes depended on the swelling degree of the polymer, which can be controlled by the time of polymer soaking in organic solvent at elevated temperature (106 °C was found to be the optimal temperature for UHMWPE). In the case of homogeneous mixtures, the resulting membranes possessed both large and small pores. They were characterized by quite high porosity (45–65% vol.), liquid permeance of 46–134 L m^−2^ h^−1^ bar^−1^, a mean flow pore size of 30–75 nm, and a very high crystallinity degree of 86–89% at a decent tensile strength of 3–9 MPa. For these membranes, rejection of blue dextran dye with a molecular weight of 70 kg/mol was 22–76%. In the case of thermoreversible gels, the resulting membranes had only small pores located in the interlamellar spaces. They were characterized by a lower crystallinity degree of 70–74%, a moderate porosity of 12–28%, liquid permeability of up to 12–26 L m^−2^ h^−1^ bar^−1^, a mean flow pore size of up to 12–17 nm, and a higher tensile strength of 11–20 MPa. These membranes demonstrated blue dextran retention of nearly 100%.

## 1. Introduction

Polymeric membranes are known to be used in water treatment [1,2], membrane crystallization [3], filtration processes for medical [4], food [5], and other industries [6], as parts of Li–ion batteries [7,8], etc. The most common membrane materials are polyvinylidene fluoride [2], polypropylene [9], polysulfone [10], polyacrylonitrile [11], and low [12] and high [13] density polyethylenes (LDPE, HDPE). Nowadays, there is a growing interest in membrane formation from ultra-high molecular weight polyethylene (UHMWPE). The latter (in comparison to conventional polyethylenes (PE)) has a higher mechanical strength [14], abrasion resistance [15], impact viscosity [16], and a greater puncture strength [17], which is a very important property for separators of Li–ion batteries. The literature analysis performed earlier [18] showed that UHMWPE membranes were mainly obtained by the sintering of powders [19] and thermally induced phase separation (TIPS) [20], including its combination with stretching [21].

In a classic variant of the TIPS method, a homogeneous solution of polymer in a solvent is first prepared at an elevated temperature. Then, it is formed into a desired shape (thin film, hollow fiber) and cooled. The cooling induces phase separation and polymer crystallization, and thus the porous structure is formed. After that, the solvent is removed from the pores of the obtained porous structure. Importantly, the variety of solvents suitable for UHMWPE membrane formation using the TIPS method is very limited due to difficulties in the preparation of homogeneous solutions. For example, in more than 80% of papers where the preparation of UHWMPE homogeneous solutions was reported, liquid paraffin and its analogs, xylene and decaline, were used as solvents. Since the selection of the solvent for TIPS is one of the ways to effectively tailor the porous structure and properties of membranes, there are fewer ways to control the structure and transport and physicomechanical properties of UHMWPE membranes. Thus, one of the trends of the last decade is the development and search for new ways to prepare porous UHMWPE-based structures [22,23,24,25,26].

For example, it has been proposed to prepare UHMWPE membranes by filtration onto non-woven support of a dispersion of UHMWPE microgels [22], by winding onto a metal frame of shish-kebab structures [23], etc. [24,25].

Previously, when we studied the thermal behavior of different mixtures of semicrystalline (SC) polymers with low molecular (LM) liquid and crystalline substances, we showed [27,28,29,30,31,32] that the cooling of swollen polymer samples or gels (solutions of LM substance in amorphous regions of SC polymer) resulted in their microphase separation. In this process, small pores filled by LM components were formed inside the amorphous regions of the polymer. Taking this fact into account, it can be assumed that swelling and subsequent deswelling phenomena can be used for membrane preparation, including UHMWPE membranes.

The phenomenon of selective swelling of block-copolymers was implemented by Wang et al. [33,34,35,36] and other researchers [37,38] to form a variety of porous structures. In the process of so-called selective swelling-induced pore generation, block-copolymer samples are immersed into a swelling agent with a high affinity to the minority block but almost inert to the majority block. The minority block regions expand due to the swelling, and then, after evaporation of the solvent (deswelling), the porous structure remains with uniform pore size and straight pore profiles.

For the individual polymers, as far as we are aware to date, the approach based on swelling and deswelling was used only in Ref. [39] for the formation of ion-selective channels in sulfonated polyether sulfone membranes. In addition, the phenomenon of pH-dependent swelling of membrane material was used to control the rejection coefficient of the membrane [40,41]. However, in the above-mentioned papers, amorphous polymers were used. Obviously, the inherent microheterogeneity (interchange of amorphous and crystalline regions) [42,43] of SC polymers, including UHMWPE, complicates the behavior of such polymers during swelling and deswelling.

Therefore, the aim of the present paper was to work out a new method of UHMWPE membrane formation based on controlled swelling and deswelling of initially monolithic films, develop its physicochemical basis, prepare the membrane samples at different swelling degrees, and assess their structure and properties.

## 2. Materials and Methods

### 2.1. Materials

Monolithic UHMWPE films kindly supplied by “AVDT” Ltd. (Moscow, Russia) were used as a raw material for membrane formation. These 155 μm thick films filled with a small amount of carbon black were prepared from Gur 4150 (Celanese Co., Irving, TX, USA) UHMWPE with a molecular weight of 9 × 10^6^ g/mol. The crystallinity degree (α) of UHMWPE in films was 55%.

O-xylene (“Chimmed” Ltd., Moscow, Russia) with a density of 0.880 g/cm^3^ at 25 °C was used as a swelling agent. Isopropyl alcohol (“Ekos-1” Co., Staraya Kupavna, Russia) with a density of 0.785 g/cm^3^ at 25 °C was used as an extractant.

### 2.2. Methods

#### 2.2.1. Differential Scanning Calorimetry

Differential scanning calorimetry (DSC) thermograms of the initial films, their blends with o-xylene, and the prepared membranes were recorded using a Phoenix 204F1 instrument (NETZSCH, Selb, Germany). The measurement protocol is shown in Figure 1.

The crystallinity degree of the polymer was calculated using the following equation:*α =* ∆*H*_m_/∆*H*_m_^100%^(1)
where ∆*H*_m_ is the melting enthalpy of the sample determined in DSC experiments, and ∆*H*_m_^100%^ = 293 J/g [44] is the melting enthalpy of the hypothetical 100% crystalline UHMWPE sample.

Taking into account the fact that melting enthalpy was calculated per unit of the full sample mass, for the mixtures of UHMWPE with o-xylene, Equation (1) was modified as follows:*α =* ∆*H*_m_/∆*H*_m_^100%^ *w*_2_(2)
where *w*_2_ is the polymer mass fraction in the mixture.

#### 2.2.2. Membrane Preparation Method

The UHMWPE membrane samples were prepared as follows. The industrially produced UHMWPE films were cut into 40 × 40 mm square samples and placed into a vial filled with o-xylene (bath modulus of 1:100). Then, the vial was heated in the oven to 106 °C and kept at this temperature for different time periods (20 to 100 min). The annealing resulted in swelling the films to different swelling degrees (Q). Subsequently, the vial with the sample was cooled to room temperature in air. After that, the film was removed from the vial and immersed in an isopropyl alcohol bath (1:100) for 3 h for extraction of o-xylene. The extractant was removed from the membrane samples by drying at 40 °C until constant mass. The conditions for preparing the five membrane samples are summarized in Table 1.

The following events occurred during the membrane preparation process. Soaking the films at an elevated temperature resulted in an increase in their volume due to swelling. Then, after cooling and transferring the films into the extraction bath, the size of the films decreased due to the extraction of solvent and shrinkage. The volume of the samples decreased even more after the removal of the extractant by evaporation but remained larger than that of the initial sample. The sizes of the initial, swollen, and dried samples were used to calculate the swelling degree (Q) and porosity (P) values using Equations (3) and (4).
Q = (*V*_s_ − *V*_o_)/*V*_o_,(3)
P = (*V*_m_ − *V*_o_)/*V*_o_,(4)
where *V*_o_, *V*_s_, and *V*_m_ are volumes of the initial, swollen, and dried samples, respectively. The volumes were calculated by multiplying the dimensions of the samples. The thickness of the initial film, and both swollen and dried samples were obtained as the average of at least 10 measurements using a micrometer in different spots. The deviation for the initial film with a thickness of 155 μm did not exceed 5 μm, while for the swollen and dried samples, it was not greater than 10 μm. Both the length and width of the samples were measured using a ruler with an accuracy of 0.5 mm. It should be noted that all the dimensions (length, width, and thickness) of the samples increased proportionally during their swelling within an experimental error.

#### 2.2.3. Measurement of Mechanical Properties of Films and Membranes

Tensile strength (σ) and relative elongation at break (ε) of the films and membranes were determined using an I-11M instrument (“Tochpribor” Ltd., Ivanovo, Russia) at an extension rate of 50 mm/min as an average of five measurements. The dimensions of the samples were 3 mm in width and 25 mm in length, so the unclamped length-to-width ratio was 5:1. Tensile strength was calculated according to Equation (5):σ = *F*_max_/*S*_o_,(5)
where *F*_max_ is the breaking force, and *S*_o_ is the area of the cross-section of the initial sample.

The relative elongation at break was calculated according to Equation (6):ε = *l*_br_/*l*_o_,(6)
where *l*_o_ and *l*_br_ are lengths of the initial sample and at break. The Young modulus was calculated at a relative elongation of 5% using Equation (7):*E* = σ/ε,(7)

#### 2.2.4. Scanning Electron Microscopy

The morphology of the surfaces and cross-sections of the samples were obtained using a Quattro S scanning electron microscope (SEM) (Thermo Fischer Scientific, Černovice, Czech Republic). The images were recorded in secondary electrons at an accelerating voltage of 5 kV. A cross-section of the samples was analyzed after freezing in liquid nitrogen. The samples for the study were coated with gold using a Quorum Q150es plus sputter coater.

#### 2.2.5. Transport Properties of Membranes

The pore size was measured by liquid-liquid displacement porosimetry using a POROLIQ 1000 ML porometer (Porometer, Nazareth, Belgium) according to the procedure described in Ref. [45]. The operating principle of this device is based on the measurement of the equilibrium pressure corresponding to the flux of the displacing liquid. The displacement of the wetting liquid was carried out by a stepwise increase in the transmembrane pressure with monitoring of the flux through the membrane after the initial stabilization time (180 s) at each applied pressure. The measurement was stopped after reaching a linear dependence of the flux on pressure, which indicated a complete displacement of the wetting liquid. The alcohol-rich phase was used as a wetting liquid, and the water-rich phase was used as a displacing liquid. The coupons (2.5 cm in diameter) were cut from the membrane and then placed into the beaker with the wetting liquid for at least 12 h at room temperature before testing. The measurements were carried out at room temperature using a pair of immiscible liquids prepared by demixing a mixture of isobutanol and water (1/4, *v*/*v*). The diameter (D) of the open pore is related to the trans-membrane pressure via the Young–Laplace equation. Two parameters were determined for each coupon: Mean flow pore size (MFP) and size of the biggest pore. Membrane permeance was also measured during these experiments.

Filtration experiments were carried out in a dead-end stirred filtration cell. For filtration experiments, the membrane coupon was placed onto porous stainless steel disks and sealed with a rubber O-ring. The active membrane area in the cell was 7.9 cm^2^. The system was pressurized with helium. The transmembrane pressure was 5 bar. Water was used as a solvent. Blue Dextran (M_w_ = 70 kg/mol) was used as a solute at a concentration of 10 mg/L. Due to the hydrophobic nature of UHMWPE, ethanol was filtered through every membrane coupon for 1 h at 5 bar to fill all membrane pores. Pure water was filtered for 2 h at 5 bar to determine pure water permeance. Then, Blue Dextran solution in water was filtered until 100 mL of permeate was collected. In the case of solution filtration, the feed was stirred at 550 rpm to minimize the concentration polarization effect. The membrane permeance *L* (L/(m^2^·h·bar)) was determined as:*L* = *m*/(*ρ∙S∙t*∙∆*p*),(8)
where ∆*p* is the trans-membrane pressure (bar), *m* is the mass of the permeate (g), *ρ* is the density of the permeate (g/L), *S* is the active membrane area (m^2^), and *t* is the filtration time (h).

The concentration of Blue Dextran in the feed and permeate was measured with a spectrophotometer at a wavelength of 620 nm. The rejection *R* was calculated using the following relation:*R* = (1 − *C_p_/C_f_*)·100%,(9)
where *C_f_* and *C_p_* denote the solute concentrations in the feed and permeate respectively.

#### 2.2.6. Measurement of Water Contact Angle of Film and Membranes

The water contact angle of the films and membranes was determined by digital analysis of the photographs of 5 μL droplets of distilled water placed on the surface of the samples. The mean value of five measurements in random places of the sample was taken as the final result.

#### 2.2.7. Evaluation of the Membranes’ Thermal Stability

To assess the thermal stability of the samples, the membranes were annealed for 1 h at different temperatures (90–120 °C) and the volume shrinkage of the membranes was calculated.

## 3. Results and Discussion

### 3.1. Theoretical Basis for the New Method of Membrane Formation

The TIPS method lies in the basis of the proposed UHMWPE membrane formation technique. General ideas on the mechanism of structure formation of the membranes during TIPS of homogeneous mixtures of SC polymers with LM liquid were proposed by Lloyd et al. [46,47]. In the development of these ideas, two important aspects of SC polymer behavior were not taken into account. One aspect is the swelling ability of SC polymers, and the other one is that the swelling is a thermoreversible process. However, in the last few years, more and more evidence appears, contributing to the fact that swelling must be accounted for in the TIPS process.

For example, in Refs. [48,49], it was reported that after cooling the HDPE mixture with liquid paraffin, the latter may reside between lamellae, clusters, or stacked clusters, depending on the mass fraction of the LM substance in the mixture. However, it was not discussed whether the liquid paraffin was dissolved in amorphous regions or formed its own phase.

In a very recent review [50] devoted to TIPS, it was stated that during cooling of the SC polymer mixtures with the solvent, the latter is “rejected” by the growing lamellae into the interlamellar spaces and thus interlamellar or interspherulitic (if the polymer crystallizes with the formation of spherulites) pores are formed. However, the reason why the liquid remains in the interlamellar spaces is explained only by kinetic factors (the relation of spherulite growth rate to the mobility of the liquid droplet). Meanwhile, in the series of papers [27,28,29,30,31,32] published before the cited review, we proposed a thermodynamic explanation for the formation of interlamellar pores. Cooling of an SC polymer mixture with an LM liquid results in polymer crystallization, which leads to the formation of the gel. In such a gel, crystallites (lamellae), connected to each other by tie chains and entangled loops, act as crosslinks of the network. The LM solvent remains dissolved in the amorphous regions of the formed gel due to the high affinity between the solvent molecules and polymer segments in the amorphous state. Cooling of the gel leads to an increase of its crystallinity degree and, consequently, a decrease of the amorphous region fraction and thus enrichment of these regions by the solvent. At the same time, cooling also results in a decrease in the solvent’s affinity for the polymer. At a certain temperature, the uniform mixture of amorphous segments of macromolecules and molecules of LM solvent becomes thermodynamically unstable. This is partly due to the decrease in thermodynamic affinity between the components and partly due to the enrichment of this mixture by the LM liquid. Such instability leads to spontaneous microphase separation of these amorphous regions, which results in the formation of small droplets of almost pure LM solvent inside the amorphous regions of the gel.

The temperature boundary at which the gel microphase separation begins is in fact a dependence of SC polymer swelling degree on temperature. This boundary curve was plotted on the temperature–composition phase diagrams for the different mixtures of SC polymers with LM substances [27,28,29,30,31,32]. One of such diagrams (for the HDPE–m-xylene system [27]), which is closely related to the mixture of UHMWPE with o-xylene discussed in the present paper, is presented in Figure 2.

The details of its construction process using an original optical method, as well as events that were realized during the heating of the mixtures of different compositions, are discussed elsewhere [27,31]. The ABC curve in this diagram is a dependence of SC polymer full amorphization (melting) on the composition of the initial mixture. The BD curve is the temperature dependence of the polymer swelling degree. The ABC and BD curves delineate three regions in the phase diagram. In region I, there is a homogeneous mixture of HDPE melt and m-xylene. In region II, uniform metastable gels (or solutions of m-xylene in amorphous regions of SC HDPE) exist. In region III, such gels coexist with pure solvent or, in the general case, with a very low-concentrated solution of a polymer in a solvent.

An analysis of this phase diagram together with data on the structural evolution of homogeneous mixtures during TIPS [30,31] and the morphology of the membranes formed after extraction of the solvent allowed us to develop a scheme of structural transformations during cooling of different mixtures (see Figure 2).

In Figure 2, one can see that for the polymer-lean homogeneous mixture (1), the crossing AB line results in the crystallization of the polymer in the form of spherulites (2). The spherulites are composed of lamellae connected by tie chains, which, together with cilia and loops, constitute amorphous regions. Since crystallites do not contain any liquid, the concentration of the latter in the amorphous regions both inside (between the lamellae) and outside the spherulites increases, which is shown by the deeper blue color. Simultaneously with polymer crystallization, the LM liquid separates into small droplets of polymer-lean liquid inside the amorphous regions in the form of small droplets (shown by blue-colored ellipses). The spherulites grow until all of the polymer outside the growing spherulites is consumed. At the same time, the continuous separation of LM liquid in the form of small droplets leads to a decrease in its concentration in amorphous regions of the polymer, as shown by the discoloration of interlamellar spaces on the scheme.

When the homogeneous mixture enriched by polymer (4) crosses the BC curve, polymer crystallization in the form of spherulites occurs. Subsequent cooling results in their growth and impingement, so that a thermoreversible gel across a whole volume of the sample is formed (5). In such a gel, the solvent is molecularly dispersed uniformly in the amorphous regions between the lamellae of the polymer. Crossing the BD curve results in the microphase separation of the formed gel so that the LM solvent separates from the amorphous regions in the form of small droplets (6).

Obviously, all of the discussed transformations, including transition 5–6, are thermoreversible. Taking this fact into account, it becomes obvious that for the preparation of porous structures via TIPS, one does not necessarily need to prepare homogeneous molecular mixtures in region I. It is sufficient to transform the SC polymer into a swollen state corresponding to region II by increasing the temperature and cooling the formed metastable gel in order to induce its microphase separation.

Therefore, the possibility of obtaining porous structures by cooling down the previously swollen samples of SC polymers provides a basis for the proposed membrane formation method.

### 3.2. Thermal Behavior of UHMWPE Mixtures with O-xylene

Unfortunately, due to very slow diffusion processes in mixtures containing UHMWPE [51], we were not able to plot the phase diagram for the UHMWPE mixture with o-xylene using the optical method since this method is quite time-consuming (up to 6 months [31]), even for mixtures containing conventional polymers. Thus, in order to investigate the thermal behavior of UHMWPE mixtures with o-xylene, we used the DSC method that allows plotting a dynamic phase diagram.

In Figure 3, the DSC-thermograms obtained during the second heating (Figure 3a) and the first cooling (Figure 3b) of UHMWPE and its mixtures with o-xylene are shown. It is seen that both endotherms and exotherms contain only one wide peak. Both the intensity and maximum temperature of this peak decrease with an increase in the o-xylene concentration in the blend. In the case of second heating, the peak reflects the combined thermal effect of two processes: dissolution of o-xylene in amorphous regions of the SC polymer (which is very small) and amorphization (melting) of the UHMWPE in the presence of o-xylene dissolved in the polymer. It should be noted that since these processes are calorimetrically indiscernible, it is impossible to plot the BD boundary curve (Figure 2) using DSC data only.

In the case of cooling the mixtures, the exothermic peak also reflects the realization of two processes: UHMWPE crystallization from its molecular mixture with o-xylene and separation of o-xylene into its own phase.

Figure 3a also shows the values of the peak area related to the sample mass in both polymer and solvent (ΔH), related to the UHMWPE mass in the samples (ΔH/*w*_2_), and calculated using Equation (2) for the crystallinity degree (α) of UHMWPE. Although the total absolute peak area expectedly decreases with a decrease in the polymer mass fraction in the sample, if the peak area is related to the polymer mass only, the opposite tendency is observed. Thus, one can see that an increase in the o-xylene concentration in the mixture leads to an increase in polymer crystallinity α from ~0.59 to 0.89. This is because the presence of o-xylene in the UHMWPE melt facilitates the kinetic mobility of the PE segments and thus kinetically favors the process of lamellae formation. At the same time, the onset of exothermic peak shifts to the lower temperatures with increasing o-xylene concentration. This means that despite the kinetic facilitation of the crystallization process, higher undercooling is needed to start crystallization in mixtures enriched by the solvent.

The maximum temperatures of the endothermic peaks were used to construct the dynamic phase diagram for the UHMWPE mixture with o-xylene. These temperatures were plotted on the earlier [27] obtained using an optical method phase diagram for the HDPE mixture with m-xylene (Figure 4). The data obtained by the DSC method for the UHMWPE–o-xylene mixture is in very good agreement with the data obtained by the optical method for the HDPE–m-xylene system. It was expected that the melting temperature of high polymers would almost be independent of the molecular weight [52] and that o-xylene and m-xylene would be very similar in terms of both chemical structure and properties.

At the same time, the location of the BD curve on the phase diagram for the UHWMPE mixture with o-xylene may be significantly different from that shown in Figure 4. As stated above, the BD curve reflects the thermodynamics of the swelling process, which is more dependent on the molecular weight of the polymer, its initial crystallinity degree, and the peculiarities of its nano-scale structure. Nonetheless, in our opinion, its location can be used as a rough approximation to estimate the equilibrium swelling degree at a certain temperature.

### 3.3. UHMWPE Membrane Preparation

As a result of preliminary experiments performed in the chosen temperature interval of 90–102 °C (region III on the phase diagram, Figure 4) based on BD curve location, it was found that the swelling kinetics at these temperatures was very slow. In particular, after 1 day of annealing, less than half of the equilibrium swelling degree (according to the BD curve) was attained. Thus, the process, on the one hand, was not effective from the technological point of view, and, on the other hand, there is a concern that during a long annealing thermal oxidative destruction of the mixture components will occur.

Therefore, we decided to carry out the swelling process of the monolithic films at higher temperatures. In another series of preliminary experiments, a temperature of 106 °C was shown to be optimal. At lower temperatures, the duration of the swelling process was still high, while at higher temperatures, the swelling was not uniform across the film. The drawback of conducting the swelling process at 106 °C i.e., in region I on the phase diagram, was a quite significant decrease of the film weight due to the thermodynamically allowed process of “washing away” (extraction) of some macromolecules from the film into the o-xylene bath. The weight loss, which was 5–20% depending on the swelling time, was taken into account in the processing obtained results.

From the phase diagram (Figure 4), it was seen that at the chosen temperature and bath modulus (1:100), the UHMWPE–o-xylene mixture must be the homogeneous solution from a thermodynamic point of view (the composition is shown by the star in Figure 4).

Based on the fact that the formation of homogeneous solutions of SC polymers is always preceded by their swelling stage [53,54,55], during which the swelling degree increases at a fixed temperature, we prepared the samples with a different swelling degree with the characteristics summarized in Table 2 by varying the swelling time. As stated in Section 2.2.2, the forming membrane suffered quite high shrinkage during the extraction and drying stages. The actual porosity of the prepared membranes was much lower than the volume fraction of the solvent in the swollen film used to prepare the corresponding membranes. It was previously shown that shrinkage of UHMWPE-based porous materials can be greatly reduced if the solvent is removed by freeze-drying [56,57] or by using supercritical CO_2_ [26] or other fluids [58] as an extractant. In the present paper, we limited ourselves to the study of samples prepared in a more industrially friendly way. Obviously, the samples that did not suffer such high shrinkage would have different properties from the ones discussed below. Their detailed investigation is the subject of the following paper.

### 3.4. Thermophysical Properties of Membranes

Figure 5a shows the DSC-thermograms obtained during the first heating of the initial film (F) and the obtained membrane samples M1–M5. Clearly, the maximum peak reflecting amorphization (melting) of UHMWPE was 137 ± 1 °C, which was almost the same for all the samples. At the same time, the peak area (absolute melting enthalpy) increased with increasing sample porosity. The α values of the membranes prepared (as shown in Table 2) from the swollen films containing 8.7–38 wt. % of the polymer was calculated using Equation (1) and plotted against the polymer mass fraction of the swollen film in Figure 5b (green solid circles). In the same figure, the dependence of the UHMWPE crystallinity degree on polymer mass fraction obtained from DSC experiments with mixtures (Figure 3a) is also shown (red solid circles and line).

From a comparison of the dependences α = *f*(*w*_2_) obtained in two series of DSC experiments, it is clear that for the mixtures depleted by the polymer (*w*_2_ ≤ 0.207), the α values of the membranes (M3–M5) and of the polymer crystallized in the DSC crucible from the homogeneous mixtures during its cooling at a rate of 10 °C/min were almost identical. At the same time, the α values of the membranes M1 and M2 were significantly lower than those of the polymer crystallized from the homogeneous mixture.

To explain the reason for such a difference of α values, we used those plotted on the phase diagram coordinates of the swollen films (polymer mass fractions in the swollen film at 106 °C) to prepare membranes M1–M5 (green points in Figure 4). Clearly, the compositions of the swollen films used to prepare membranes M3–M5 reside in region I (homogeneous mixtures) of the phase diagram, while those of the films used to prepare membranes M1 and M2—in region II (swollen SC polymers—metastable gels with crystallites acting as crosslinks). Thus, the crystalline structure in membranes M1 and M2 was restricted by the existing (incompletely destroyed) network conditions, and thus, the attained α values were lower than those attained if the crystallization occurred with the homogeneous mixtures of similar composition.

One more piece of evidence in favor of the abovementioned is that the shape of the thermograms of the membrane samples M1 and M2 was similar to that of the initial film, while the shape of the membranes M3–M5 thermograms was clearly different (Figure 5a).

### 3.5. Mechanical Properties of Membranes

Figure 6 shows examples of the engineering stress-strain curves (Figure 6a) of the film and the membrane samples both in longitudinal (LD) and transverse (TD) directions and the mean values of the tensile strength (Figure 6b) and relative elongation at break (Figure 6c) of these samples. was is seen that the σ and ε values of the film in the longitudinal direction (LD) were ~1.25 and 1.6 times higher than those in the transverse direction (TD). This means that the film obtained by skiving from the cylindrical block had some degree of anisotropy of the mechanical properties.

As expected, the σ and ε values of the porous samples were lower than those of the monolithic film. The tensile strength continuously decreased with increasing porosity, while anisotropy disappeared for the membrane samples M3–M5 (both σ and ε values were almost the same in LD and TD for these samples). At the same time, while ε in LD decreased continuously with increasing porosity of the sample, ε measured in TD passed through a minimum in the M1–M5 series. As noted above, the ε values in LD and TD were the same for the M3–M5 membranes. This was a result of the erase of the thermal history (erase of anisotropy of the initial structure) of the films due to their transition in region I (Figure 4), where homogeneous mixtures of UHMWPE and o-xylene existed. On the contrary, in samples of M1 and M2, obtained from films swollen to a gel state (region II in Figure 4), the initial structure, which led to the anisotropy of mechanical properties, was still preserved due to the presence of crystallites acting as network crosslinks.

The character of variation of the ε values measured in TD means that the increase of porosity of the samples formed from the gels that kept the memory of the initial anisotropy of the film resulted in acceleration of the defect formation in this direction. It should be noted that ε (which characterizes the defectiveness of the structure) of more porous M3–M5 membranes obtained from homogeneous mixtures was even lower than that of less porous M1 and M2 membranes but obtained from gels that retained the memory of the initial anisotropic structure.

It should be noted that Figure 6 shows the engineering stress–strain data that did not account for (1) a decrease in the sample cross-section area in the course of their stretching and (2) the fact that in membrane samples a part of the cross-section area was filled by pores. To account for these comments, Equation (5) was modified as follows:σ_true_ = *F*_max_ (1 + ε)/(1 − P) *S*_o_,(10)

In Table 3, the mean values of the engineering σ and ε, true σ values, and Young moduli of the initial film and membrane samples are summarized.

From the analysis of Table 3, it can be concluded that the difference between σ_true_ in LD and TD of the initial film and membrane samples M1 and M2 was even more pronounced. Furthermore, the σ_true_ values in TD were similar to the discussed above variation of ε; true strength of the less porous membranes M1 and M2 in TD was even lower than that of the more porous M3 and M4 membranes. Again, this is due to different mechanisms of structure formation in the M1, M2, and M3–M5 samples. Young’s modulus expectedly decreases with increasing sample porosity.

The observed change in the mechanical properties of the membranes, depending on their porosity, confirms the conclusion drawn above: the effect of the physical state of the mixture on the formation of the crystalline structure during cooling.

### 3.6. Morphology of Membranes and Film

Figure 7 shows the SEM images of the surface of the initial film (F) and membrane samples M1–M5. Clearly, the surface of the initial film was generally uniform but contained some defects. Bearing in mind that their preparation method was based on skiving the film from a cylindrical block and taking into account their characteristic shape, it can be assumed that the defects were associated with the wear of the equipment used to obtain the films. All the membranes, including those with low overall porosity (see Table 2), had a quite porous surface that contained both large pores with a size of several tens of microns (see images in left and middle columns) probably located in the defects of the initial film and small pores with a size up to 200 nm, uniformly distributed across the surface. The defects, however, did not propagate all the way through the thickness of the sample and thus did not affect the overall transport properties of the membranes. From the images taken at low magnification, it was also clear that an increase in overall porosity led to a decrease in the relative area of the smooth almost non-porous surface, which appeared darker in the images in the left column. Nonetheless, on scales of several hundreds of microns, the pores were distributed uniformly across the surface.

Figure 8 shows the morphology of the cross-sections of the initial film and membrane samples M1–M5. Although the initial film was monolithic, the cross-section appeared a little porous due to the plastic deformation of UHMWPE, which was realized even at −196 °C (the fracture was ductile even in liquid nitrogen medium) [59]. Analysis of the cross-section morphology of the membranes showed that despite the gradient conditions of the swelling process (from the surfaces of the film to its center), there was almost no anisotropy in the porous structure across the thickness of the membrane (see the left column in Figure 8), i.e., the pores were distributed quite uniformly. The apparent porosity of the membrane samples M1 and M2 was higher than their actual values (Table 2) due to the ductile character of the sample fracture. Nevertheless, the size of separate pores in these samples did not exceed several hundreds of nm, which agreed well with the described above mechanism of their formation via microphase separation of the swollen SC polymer (separation of liquid into a separate phase in the form of small droplets from the amorphous regions of SC polymer).

The microstructure of samples M3–M5 was substantially different from that of the M1 and M2 membranes. Although these samples also contained features of ductile fracture, it was seen that the pore size in these membranes was as high as several microns. Taking into account the fact that such pores could not be the result of the microphase separation of the gel, it can be concluded that such pores arose as a result of polymer crystallization from the homogeneous molecular mixture of the components. The formed structure in membrane samples M3–M5 could be characterized as leafy [24,58,60]. Interestingly, taking into account the very high values of α (Table 2), the observed thickness of leaves, and the data obtained using SAXS, TEM, and other methods [61,62,63] concerning the average thickness of UHMWPE crystallites, it could be concluded that some (the thinnest ones) leaves were individual lamellar crystallites covered with thin amorphous “coat” of loops and cilia. The other leaves must be clusters containing no more than 3 lamellae separated by very thin amorphous layers.

From Figure 8, it is also clear that the porosity of the samples increased in series M1–M5. The membrane samples M4 and M5 also contained regions of ca. 50–100 microns in size, with the structure significantly different from the main structure of the sample (see the bottom two images in the right column). These regions contained “shish-kebab” structures that were obviously absent in the initial structure of the film. These structures, as well as leafy morphology, arose due to the crystallization of UHMWPE directly from the homogeneous mixture with o-xylene. Probably due to the non-uniformity of the swelling processes, the polymer concentration in these regions was a little bit lower than that in the bulk of the film and thus crystallization of polymer in a form of “shish-kebab” structures was favored.

The SEM data shown in Figure 7 and Figure 8 is a direct confirmation of the fact that the structures forming in the membranes and, as shown above, peculiarities of the polymer crystallization are indeed the functions of the physical state of the mixtures prior to their TIPS. In particular, these data showed that the M1 and M2 samples were obtained from gels (swollen but still containing initial crystallites SC UHMWPE), while the M3–M5 samples were formed from homogeneous mixtures of components.

### 3.7. Transport Properties of Membranes

The results of the transport properties measurements, water contact angle, and thermal stability of the initial film and membranes are summarized in Table 4. Figure 9 shows the MFP (Figure 9a) and permeance (Figure 9b) values plotted as a function of the sample’s overall porosity of the membranes. Unexpectedly, even the samples with very low overall porosity (12% for M1) were permeable for isobutanol, i.e., there were through pores in this sample. Figure 10 also shows that both MFP and permeance of the membranes predictably increase with increasing sample porosity. Importantly, the rate of these characteristics’ increase was higher for the M3–M5 samples than for the M1 and M2 samples. This is another confirmation of the fact that the forming structure depends on the physical state of the mixture before its cooling and is probably related to the different continuity of pores in the samples formed from gels and homogeneous mixtures.

According to the values of MFP and permeance, it could be concluded that the obtained membranes are suitable for ultrafiltration processes. Filtration of Blue Dextran solution demonstrated high rejection for membranes M1 and M2. Analysis of permeate samples after filtration through such membranes showed that the dye content was lower than random measurement errors. A further increase in membrane porosity and pore size led to a rejection decrease down to 22% for membrane M5.

The measurements of the water contact angle of the initial film and membranes showed that the membranes were, as expected, hydrophobic. The transition from monolithic film to porous samples resulted in a slight increase in the water contact angle due to an increase in surface roughness.

The experiments aimed at the evaluation of thermal stability showed that both the initial film and the membrane samples with low porosity (M1, M2) did not shrink after 1 h of annealing, even at 120 °C. The most porous samples, M4 and M5, shrank slightly after annealing at 100 °C. The maximum value of volume shrinkage did not exceed 5–6% at 120 °C. This data allows concluding that the obtained membrane samples are quite thermostable.

### 3.8. Mechanism of Structure Formation of Membranes Prepared by the Controlled Swelling/Deswelling Method

In conclusion, taking into account the obtained data on the morphology, thermophysical, and other properties of the membranes obtained by the developed method, let us consider the mechanism of their structure formation using the scheme shown in Figure 10. The initial film, even at elevated temperature (point 1), is an SC polymer sample containing both amorphous and crystalline regions, as shown in circle 1. Annealing of this film in a solvent bath results in an increase in the film volume due to swelling and a decrease of its crystallinity degree as the solvent penetrates into amorphous regions of the film. During swelling, the film composition moves to point 2. If the swelling process is terminated at point 2 (for example, by cooling the swollen film with the structure shown in circle 2), new crystallites may arise in the place of destroyed ones, the volume fraction of amorphous regions will decrease, and the local concentration of solvent in the amorphous regions will increase. Cooling also results in a continuous decrease in thermodynamic affinity between the polymer and the solvent. Thus, when the temperature corresponding to the BD curve is attained, microphase separation of the gel leads to the formation of small droplets of solvent inside the amorphous regions of the polymer. The system, now containing small droplets of solvent in amorphous (interlamellar) spaces, is shown schematically in circle 3.

In a way, the pore formation mechanism is very similar to the selective swelling-induced pore generation in block-copolymers [33,34,35,36,37,38]. In the case of UHMWPE homopolymer, the crystalline regions of the polymer act as impermeable blocks, while the solvent selectively swells the amorphous regions. The difference, however, is that both crystalline and amorphous regions of the SC polymer are different only in terms of their physical state, not their chemical structure. Thus, the transformation of one into the other is possible and their ratio changes in both the swelling and deswelling process. In the swelling stage, the crystalline regions are destroyed and transformed into amorphous regions due to the temperature and swelling pressure produced by the solvent. Then, in the deswelling stage, some amount of the polymer in the amorphous regions crystallizes as the temperature decreases and the solvent is removed.

If the swelling of the system at point 2 is continued, the polymer continues to swell and finally transforms into a liquid molecular mixture of polymer melt and solvent (point 4 in region I on the phase diagram) schematically shown in circle 4. If the process is stopped by cooling the system located at point 4, the polymer will start to crystallize (after crossing the AB line) in the mixture, still retaining its shape due to the very high quantity of intermacromolecular entanglements. As the polymer crystallizes, the macromolecules are “pulled” to the growing nuclei, and the porous structure forms (shown schematically in circle 5). This structure has a degree of crystallinity higher than that of the initial film and contains both large (between the stacks of lamellae) and small (interlamellar) pores.

In this case, the mechanism of structure formation is not different from the standard TIPS mechanism [18,31,32,40,41]. However, the proposed method could be very useful if, for some reason, it is difficult either to obtain a homogeneous mixture of the polymer and the solvent or to form it into the desired shape, which is the case for UHMWPE because of the very high viscosity of its solutions.

Thus, membranes M1 and M2 formed according to scenario 1–2–3, while the membranes M3–M5 formed according to scenario 1–4–5. The difference in these scenarios explains the difference in morphology, crystallinity degree, and mechanical and transport properties of the obtained membranes.

## 4. Conclusions

A new method for preparing UHMWPE membranes for ultrafiltration was proposed. This is based on an analysis of the topology of the phase diagram for the UHMWPE mixture with o-xylene. In this method, industrially produced monolithic films were first subjected to controlled swelling at elevated temperature. Then, cooling of the swollen films resulted in TIPS, which led to the formation of the porous structure. Finally, the solvent was removed by extraction, and the extractant was removed by drying to produce a porous membrane.

The membrane samples were obtained with porosity in the range of 12 to 65% vol. The morphology, thermal stability, and thermal, mechanical, and transport properties of the obtained samples were investigated. It was shown that depending on the attained swelling degree, the samples could be divided into two groups. The samples referring to the first group were prepared from homogeneous mixtures of UHMWPE with o-xylene obtained as a result of unrestricted swelling. These samples were characterized by a leafy structure, a very high crystallinity degree, high porosity, and permeance. For these samples, the membrane rejection of Blue Dextran Dye with a molecular weight of 70 kg/mol was 22–76%. The samples referring to the second group were prepared from the initial network (crystallites connected by tie-chains and entangled loops) gels that also formed due to swelling. These samples were characterized by small pores, a lower crystallinity degree, and permeability. These membranes demonstrated nearly 100% rejection of Blue Dextran. The mechanical strength of the obtained membranes ranged from 3.2 to 20 MPa.

Taking into account all the data on the morphology and properties of the membranes, a structure formation mechanism was proposed.

## Figures and Tables

**Figure 1 membranes-13-00422-f001:**
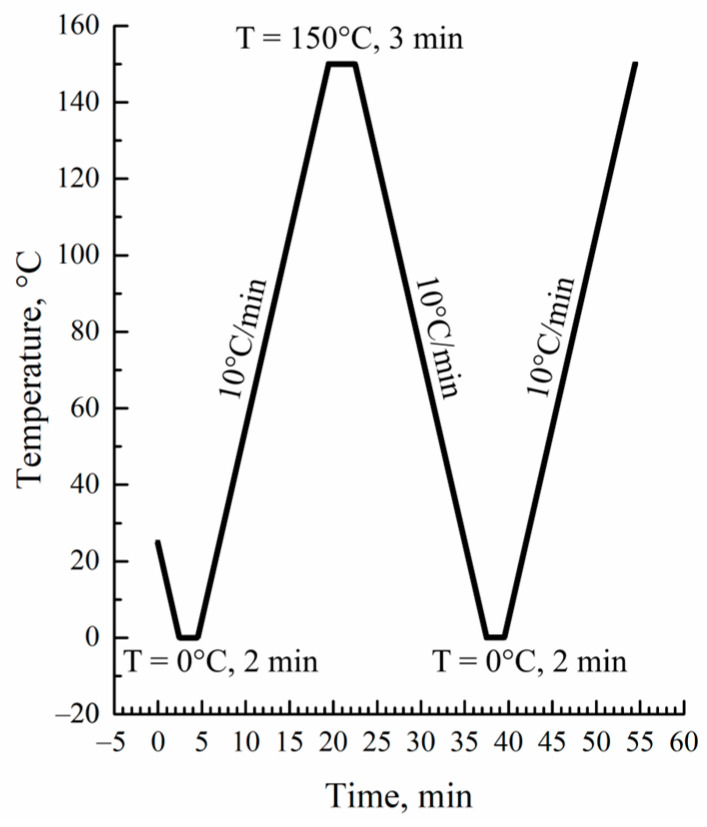
DSC measurement protocol.

**Figure 2 membranes-13-00422-f002:**
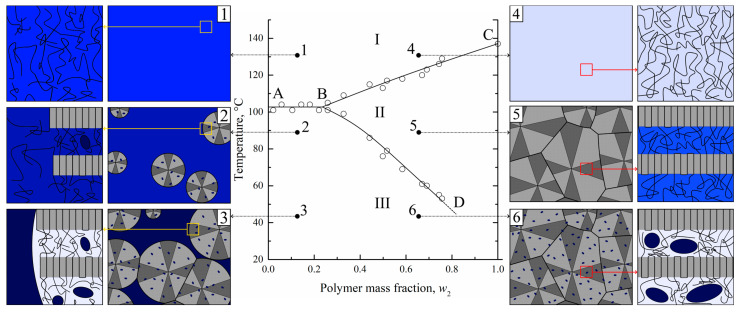
The phase diagram for the HDPE–m-xylene mixture [27] constructed by optical method and schemes [31], illustrating structure evolution on a micro- and nanoscale during cooling of the solvent-enriched (1–3) and polymer-enriched (4–6) homogeneous mixtures. Blue color intensity stands for the solvent concentration. Adapted with permission from L.N. Mizerovskii, T.N. Lebedeva, K.V. Pochivalov, Polymer Science Series A; Published by Springer-Nature, 2015 and K.V. Pochivalov et al. Materials Today Communications; Published by Elsevier, 2021.

**Figure 3 membranes-13-00422-f003:**
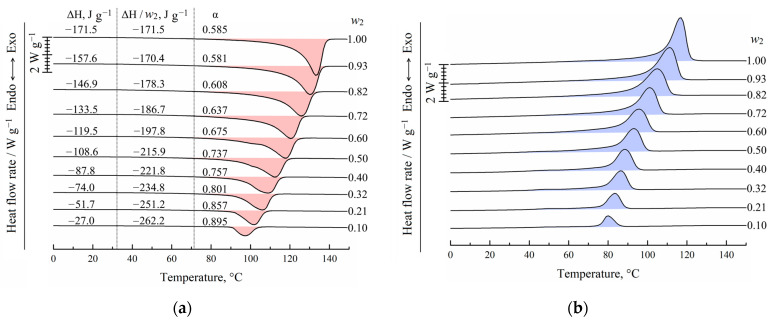
DSC-thermograms for the UHMWPE and its mixtures with o-xylene obtained during (**a**) second heating and (**b**) first cooling. The mass fraction of the polymer (*w*_2_) is shown to the right of the curves. In (**a**), peak area (ΔH), melting enthalpy of UHMWPE (ΔH/*w*_2_), and corresponding crystallinity degree (α) values are shown for each curve.

**Figure 4 membranes-13-00422-f004:**
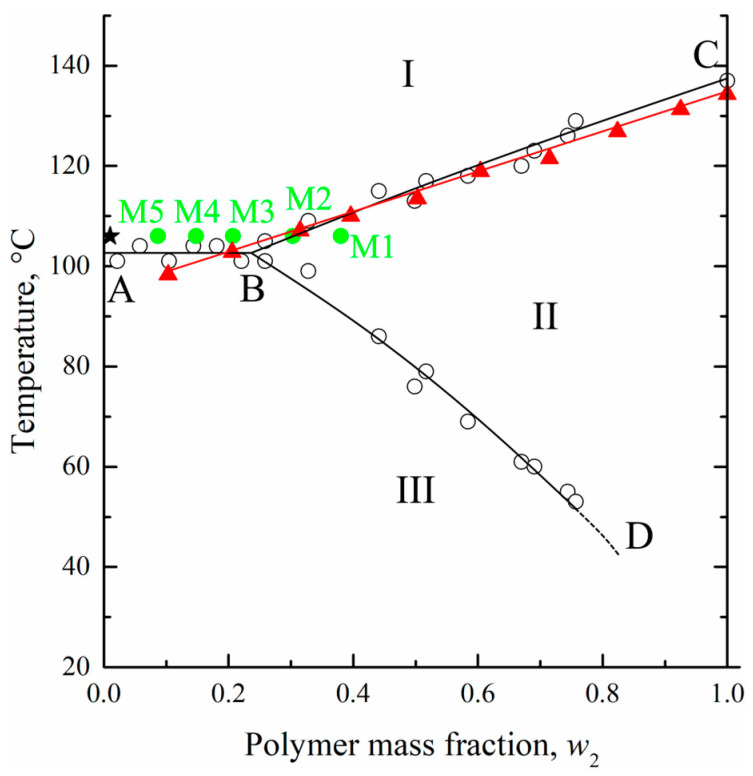
The phase diagram for the HDPE–m-xylene mixture constructed by the optical method [27] (hollows circles and black curves) together with DSC data for the UHMWPE–o-xylene mixture (red triangles and curve). See explanation of star and green points in the text. Adapted with permission from L.N. Mizerovskii, T.N. Lebedeva, K.V. Pochivalov, Polymer Science Series A; Published by Springer-Nature, 2015.

**Figure 5 membranes-13-00422-f005:**
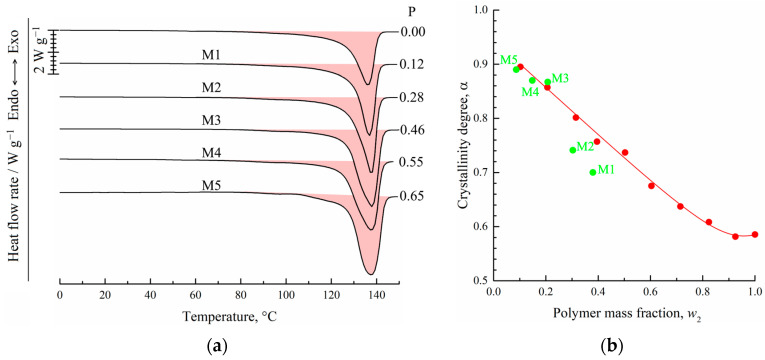
DSC-thermograms obtained during the first heating of the film and membrane samples M1–M5 (**a**). The porosity of the samples (P) is shown to the right of the curves. Dependence of the UHMWPE crystallinity degree on polymer concentration in its initial mixture with o-xylene (**b**). Red points were plotted according to DSC data for mixtures during their second heating. Green points correspond to values of the crystallinity degree of the membrane samples M1–M5.

**Figure 6 membranes-13-00422-f006:**
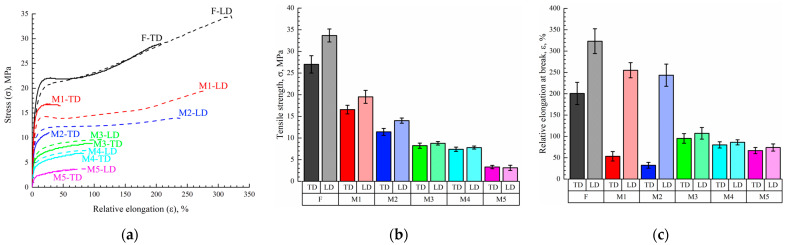
(**a**) Examples of engineering stress-strain curves of the initial film and membrane samples both in longitudinal (LD, dashed curves) and transverse (TD, solid curves) directions and mean values of (**b**) tensile strength and (**c**) relative elongation at break of these samples.

**Figure 7 membranes-13-00422-f007:**
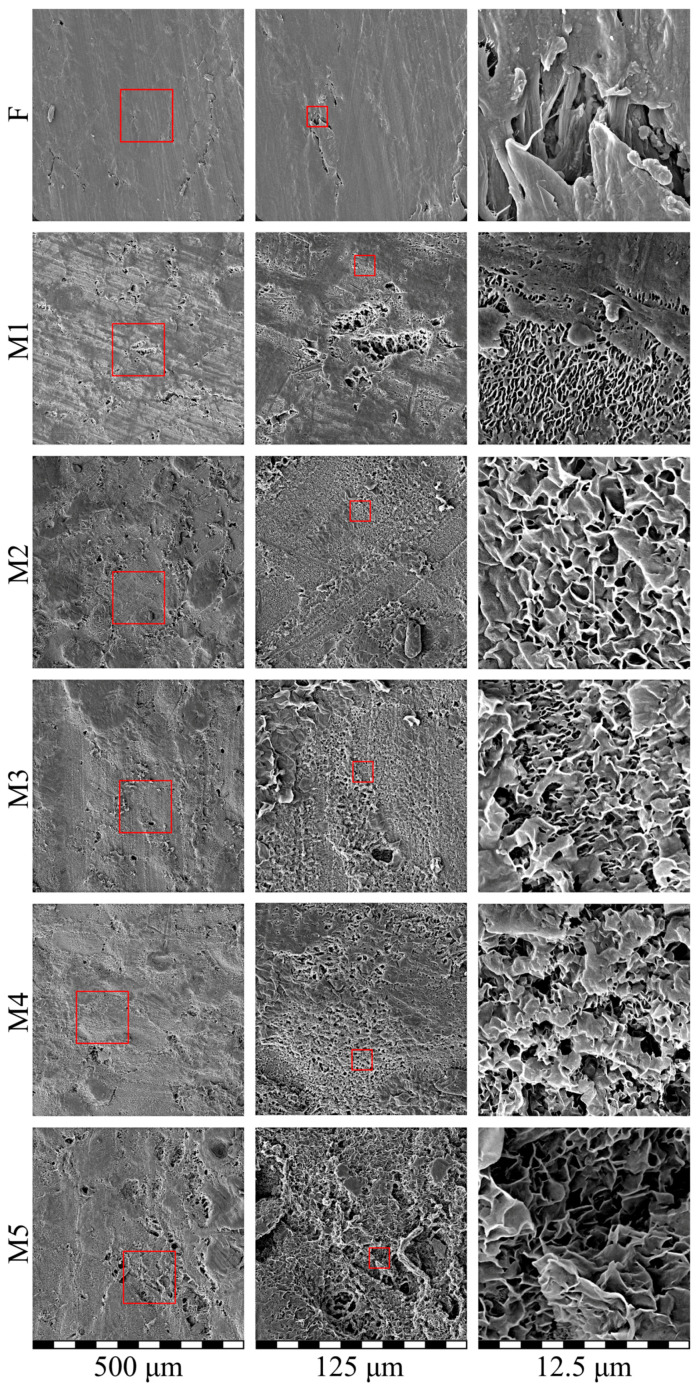
SEM images of the surface of the initial film (F) and membrane samples (M1–M5). Red squares highlight the regions where higher magnification images (shown to the left) were taken.

**Figure 8 membranes-13-00422-f008:**
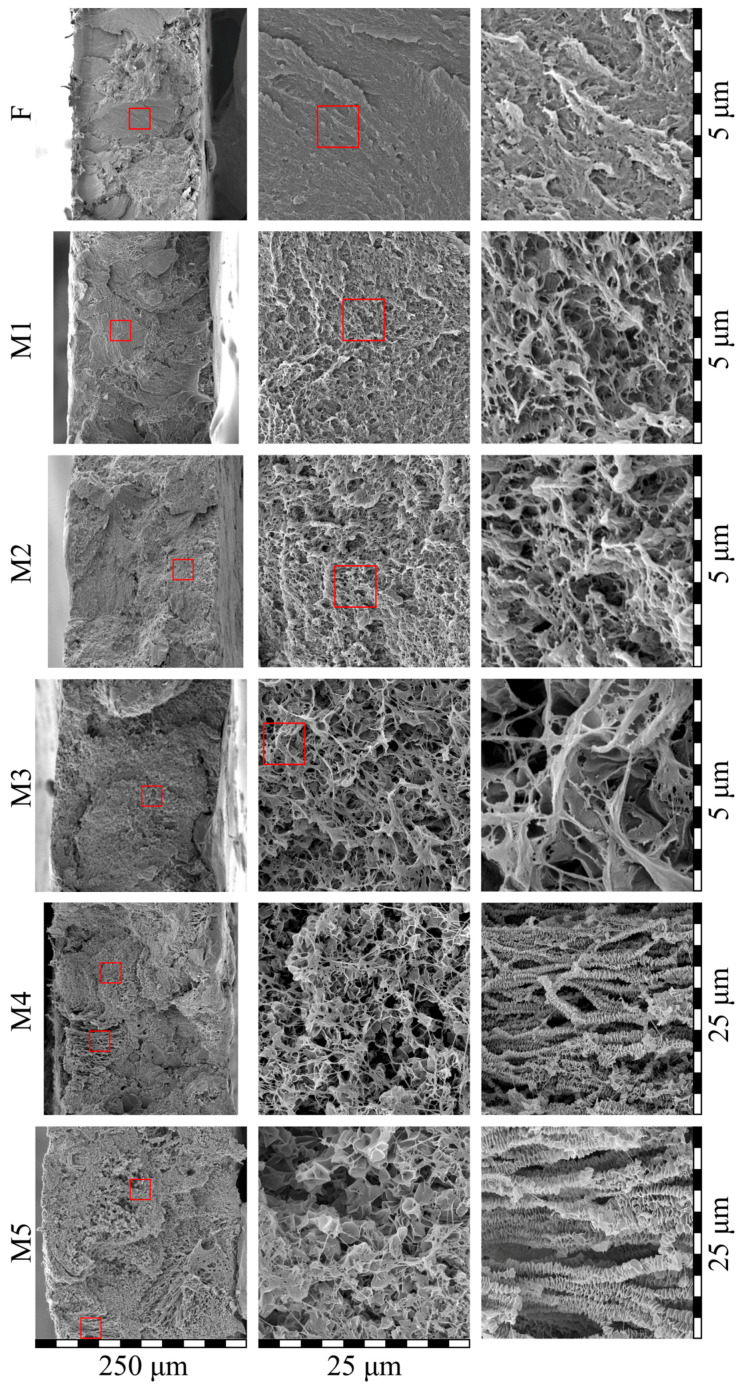
SEM images of the cross-section of the initial film (F) and membrane samples (M1–M5). Red squares highlight the regions where higher magnification images (shown to the left) were taken.

**Figure 9 membranes-13-00422-f009:**
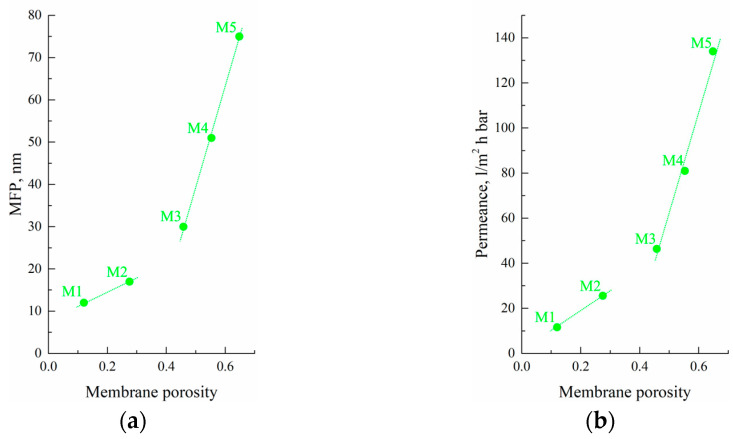
(**a**) MFP and (**b**) permeance as functions of the porosity of the obtained membrane samples.

**Figure 10 membranes-13-00422-f010:**
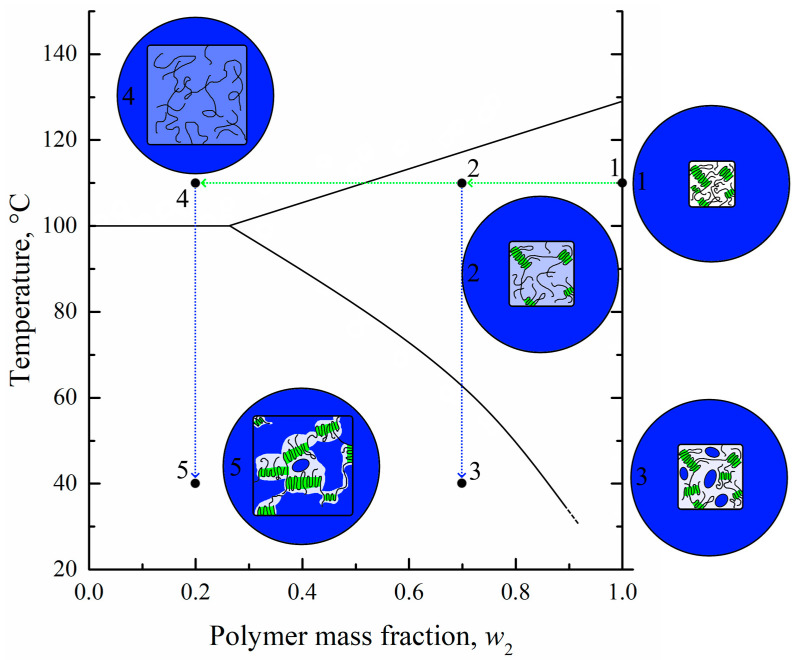
Scheme illustrating the mechanism of the structure formation of membranes by the proposed controlled swelling/deswelling method. Blue color reflects solvent; UHMWPE crystallites are colored green; white and light blue color reflects pure (without solvent) and swollen amorphous regions of UHMWPE, respectively.

**Table 1 membranes-13-00422-t001:** The parameters of the membrane preparation process.

Sample	Swelling Time, min	Swelling Temperature, °C	Swelling Degree (Q)	Volume Fraction of O-xylene in Swollen Film	Mass Fraction of UHMWPE in the Swollen Film
M1	20	106	1.7	0.630	0.380
M2	35	106	2.4	0.706	0.303
M3	45	106	4	0.800	0.207
M4	60	106	6	0.857	0.148
M5	100	106	11	0.917	0.087

**Table 2 membranes-13-00422-t002:** Some parameters of the membrane preparation process: porosity and crystallinity degree of the obtained membranes.

Sample	Swelling Time, min	Swelling Degree (Q)	Volume Fraction of O-xylene in Swollen Film	Volume Porosity (P)	Mass Fraction of UHMWPE in the Swollen Film	Crystallinity Degree (α)
M1	20	1.7	0.630	0.120	0.380	0.7
M2	35	2.4	0.706	0.275	0.303	0.741
M3	45	4	0.800	0.458	0.207	0.867
M4	60	6	0.857	0.553	0.148	0.87
M5	100	11	0.917	0.648	0.087	0.89

**Table 3 membranes-13-00422-t003:** Mechanical properties of the initial film (F) and resulting porous membranes (M1–M5).

Sample	F		M1		M2		M3		M4		M5	
Direction	TD	LD	TD	LD	TD	LD	TD	LD	TD	LD	TD	LD
σ, MPa	27	33.7	16.6	19.5	11.4	14	8.2	8.8	7.4	7.8	3.3	3.1
ε, %	200	323	53	255	32	243	95	107	80	86	67	74
σ_true_, MPa	81.2	143	28.9	78.7	20.8	66.3	29.6	33.5	29.8	32.4	15.6	15.4
E, MPa	303	275	287	240	160	175	109	125	92	98	37	33

**Table 4 membranes-13-00422-t004:** The results of measurements of transport properties, water contact angle, and thermal stability of the initial film and membranes.

Sample		F	M1	M2	M3	M4	M5
The biggest through pore, nm		–	62	82	142	240	259
MFP, nm		–	12	17	30	51	75
Permeance, L m^−2^ h^−1^ bar^−1^		–	11.6	25.6	46.4	81	134
Rejection, %		–	>99	>99	76	53	22
water contact angle, °		125	128	128	131	129	130
Volume shrinkage (%) at a temperature	90 °C	0	0	0	0	0	0
	100 °C	0	0	0	0	0.9	1.3
	110 °C	0	0	0	0.7	3.1	5.2
	120 °C	0	0	0	1.5	5.1	5.8

## Data Availability

The data presented in this study are available on request from the corresponding author.

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
