# Peer review of "Controlled Swelling of Monolithic Films as a Facile Approach to the Synthesis of UHMWPE Membranes"

_membranes, 2023, doi:10.3390/membranes13040422_

Round 1
Reviewer 1 Report
It is an interesting way to prepare UHMWPE membranes by swelling monolithic films, I think the paper could be accepted, before minor revision. Here are some advices:
1. How to measure the volumes of membranes (line 122-130), because the monolithic film (thickness 150 μm) or membrane is thin, I think it is difficult to obtain the accurate volume.
2. In section 2.2.6 (line 168) the volume of droplets, maybe 5μL was miswritten as 50μL.
3. Most UHMWPE membranes are used for microfiltration processes, while this method could obtain ultrafiltration membranes, not only the permeability but also selectivity should be discussed in this paper, and the comparison with other membranes should also be discussed.
4. I want to see the digital photos of the dried membrane, does it become wrinkled after drying? Especially for M5 (swelling for 100 min).
Author Response
Response to Reviewer 1 Comments
Point 1: How to measure the volumes of membranes (line 122-130), because the monolithic film (thickness 150 μm) or membrane is thin, I think it is difficult to obtain the accurate volume.
Response 1: The volumes of the samples were calculated by multiplying their dimensions. Since the initial film is industrially produced, its thickness was quite uniform (error less than ±5μm). Thickness values of initial, swollen and dried samples were obtained as the average of at least 10 measurements using a micrometer in different spots. The length and width of the samples were measured using a ruler with an accuracy of 0.5 mm. It should be noted that all the dimensions (length, width and thickness) increased proportionally within experimental error. We have added this information into the paper text. Although we agree that the volume obtained this way is not very accurate, it was enough to assess swelling degrees and porosities.
Added text 1: The volumes were calculated by multiplying the dimensions of the samples. The thickness of the initial film, and both swollen and dried samples was obtained as the average of at least 10 measurements using a micrometer in different spots. The deviation for the initial film with a thickness of 155μm did not exceed 5 μm, while for the swollen and dried samples it was not greater than 10 μm. Length and width of the samples were measured using a ruler with an accuracy of 0.5 mm. It should be noted that all the dimensions (length, width and thickness) of the samples increased proportionally during swelling within experimental error. (In Section 2.2.2)
Point 2: In section 2.2.6 (line 168) the volume of droplets, maybe 5μL was miswritten as 50μL.
Response 2: Thank you for your careful reading. Indeed, it is a misprint and it should be 5 μL. The text is corrected accordingly.
Point 3: Most UHMWPE membranes are used for microfiltration processes, while this method could obtain ultrafiltration membranes, not only the permeability but also selectivity should be discussed in this paper, and the comparison with other membranes should also be discussed.
Response 3: Thank you for the suggestion. We have introduced the data on selectivity of the membranes into the text.
Added text 3: Filtration experiments were carried out in a dead-end stirred filtration cell. For filtration experiments, the membrane coupon was placed onto the porous stainless steel disks and sealed with a rubber O-ring. The active membrane area in the cell was 7.9 cm2. The system was pressurized with helium. The transmembrane pressure was 5 bar. Water was used as a solvent. Blue Dextran (Mw=70 kg/mol) was used as a solute with a concentration of 10 mg/L. Due to the hydrophobic nature of UHMWPE ethanol was filtered through every membrane coupon for 1 hour at 5 bar to fill all membrane pores. Pure water was filtered for 2 hours at 5 bar to determine pure water permeance. Then Blue Dextran solution in water was filtered until 100 ml of permeate was collected. In the case of solution filtration, the feed was stirred at 550 rpm to minimize the concentration polarization effect. The membrane permeance L (L/(m2·h·bar)) was determined as:
|
L=m/(ρ∙S∙t∙∆p), |
(8) |
where ∆p is the trans-membrane pressure (bar), m is the mass of the permeate (g), ρ is the density of the permeate (g/L), S is the active membrane area (m2), and t is the filtration time (h).
The concentration of Blue Dextran in the feed and permeate was measured with a spectrophotometer at the wavelength of 620 nm. The rejection R was calculated using the relation:
|
R=(1-Cp/Cf)·100%, |
(9) |
where Cf and Cp denote the solute concentrations in the feed and permeate respectively. (In Section 2.2.5)
Filtration of Blue Dextran solution demonstrated high rejection for membranes M1 and M2. Analysis of permeate samples after filtration through such membranes showed that dye content was lower than random measurement errors. Further increase in membrane porosity and pore size leads to a rejection decrease down to 22% for membrane M5. (In Section 3.7)
Point 4: I want to see the digital photos of the dried membrane, does it become wrinkled after drying? Especially for M5 (swelling for 100 min).
Response 4: That is a very good question. Indeed, since the membranes underwent quite high shrinkage, their surfaces became a little wrinkled after drying as you may notice from the pictures provided in the attachment (the membranes are placed on a millimeter-scale paper). Please note that the main goal of the paper was to demonstrate that soaking of the dense films (available on the market) in solvent can be used to prepare porous membranes. Successful further implementation of this approach as well as search for the ways to improve it (including ways to reduce shrinkage and to increase quality of membranes) requires additional work and optimization of fabrication protocol as stated in line 388-394 of the revised unmarked manuscript.

Reviewer 2 Report
This work discussed a method for prepare porous PE membranes by swelling. It is interesting. Before acceptance, some question should be addressed.
1) How do you measure the volumes of initial, swollen, and dried samples ? As for M5, Vs/Vo =12, which meaned the very clear volume increase.
2) Fig2, Fig5 and Fig11 contain the same curve data. It should be adjusted.
3) Did you record the weight loss in the membrane preparing process? Does it made the pores?
4) In abstract, the data of the optimal membrane can be provided, not the best levels for five membranes.
5) The unit of flux (l/m2 h bar) is bad format.
Author Response
Response to Reviewer 2 Comments
Point 1: How do you measure the volumes of initial, swollen, and dried samples ? As for M5, Vs/Vo =12, which meaned the very clear volume increase.
Response 1: The volumes of the samples were calculated by multiplying their dimensions. Thickness values of initial, swollen, and dried samples were determined in several spots using a micrometer, while their length and width were measured using a ruler. It should be noted that all the dimensions (length, width and thickness) increased proportionally within experimental error. We have added this information into the paper text. Thus for the sample with the highest swelling degree the dimensions of the membrane increased about cubic root of 12 i.e. ~2.3 times. For dried samples thickness is clearly seen in SEM images of the cross-section and lies in good agreement with actual porosity.
Added text 1: The volumes were calculated by multiplying the dimensions of the samples. The thickness of the initial film, and both swollen and dried samples were obtained as the average of at least 10 measurements using a micrometer in different spots. The deviation for the initial film with a thickness of 155μm did not exceed 5 μm, while for the swollen and dried samples it was not greater than 10 μm. Both length and width of the samples were measured using a ruler with an accuracy of 0.5 mm. It should be noted that all the dimensions (length, width and thickness) of the samples increased proportionally during their swelling within an experimental error. (In Section 2.2.2)
Point 2: Fig2, Fig5 and Fig11 contain the same curve data. It should be adjusted.
Response 2: Figure 2 contains only the phase diagram for the high density polyethylene mixture with m-xylene reprinted from our previous papers and schematic snapshots of the mixture structure in different regions. In Figure 5 optical data for the high density polyethylene mixture with m-xylene is compared to the DSC data obtained for the used in the present paper UHMWPE mixture with o-xylene. Also, in Figure 5 the compositions achieved during swelling (as a first step in membrane preparation process) are shown. Lastly, Figure 11 shows schematic phase diagram for any mixture of semicrystalline polymer with good solvent to illustrate the speculation on mechanism of pore formation. Although all of these data could, in principle, be combined in one figure, in our opinion such figure would be overloaded and hard to read. Thus we think that such recurrence of the phase diagram with different added data and accent serves better readability.
Point 3: Did you record the weight loss in the membrane preparing process? Does it made the pores?
Response 3: Thank you for the question. As stated in lines 370-371 (please see revised unmarked version) of the paper, a weight loss of ca. 5-20% was recorded depending on the swelling time. The recorded weight loss was taken into account during the estimation of swelling degree and porosity. However, such partial extraction of the polymer did not play a major role in formation of pores. In this case extraction of some macromolecules is not similar to extraction of soluble component from the insoluble matrix since the extracted macromolecules do not leave empty space in the sample. Instead local microregions with lower polymer concentration are formed due to extraction. Pores in the samples (both in the bulk and local regions with lower polymer concentration) are formed due to thermally induced phase separation of either homogeneous mixture (M1–M3) or swollen semicrystalline film (M4, M5).
Point 4: In abstract, the data of the optimal membrane can be provided, not the best levels for five membranes.
Response 4: Thank you for the comment. The abstract was adjusted.
Point 5: The unit of flux (l/m2 h bar) is bad format.
Response 5: The permeance unit is changed to L m–2 h–1 bar–1.
Reviewer 3 Report
In this work, the authors reported a new method of fabrication of porous membranes based on UHMWPE by controlled swelling of the dense film. It was shown that the porous structure and filtration performance of the membranes de pended on the swelling degree of the polymer. In the case of homogeneous mixtures, the resulting membranes possessed both large and small pores as well as quite high porosity (up to 65 vol.%), while the membranes formed in the case of thermoreversible gels exhibited only small pores located in the interlamellar spaces with a lower porosity of 28%. The authors also tried to explain the mechanism of the structure formation in UHMWPE membranes. However, I suggest the following concerns should be addressed properly before considering this work for publication
1. There is an extensively studied method based on swelling to produce UF membranes, which is termed as "selective swelling-induce pore generation" of block copolymers. See Ref. Acc. Chem. Res., 2016, 49, 1401. This should be included in the Introduction part.
2. There were no rejection results; these should be added to confirm thus-produced membranes were really UF membranes as the authors claimed.
3. The porous regions on the swelling-treated films were present scarcely and randomly. The reviewer thinks the pores may be caused by the dissolution of some low-molecular-weight PE. This possibility should be discussed. Pls compare the weight of the sample before and after swelling. If the weight is reduced after soaking, the dissolution occurs.
4. Despite the emphasis of swelling of UHMWPE dense films, this method is in fact based on the thermal induced phase separation (TIPS), but the authors did not clearly distinguish their work with normal TIPS. The authors should explicitly emphasize the superiority of their work than TIPS, otherwise they should not propose such a new method.
5. The authors mentioned in line 450 “... probably located in the defects of initial film....”, so it is meaningless to measure the permeance or evaluate the permeance as functions of porosity because the defects are disparate for each membrane.
6. In line 128, the swelling degree (Q) and porosity (P) values were calculated using volumes of initial, swollen, and dried samples, but the authors did not introduce how to test the volumes of samples.
7. The authors claimed this work is based on the swelling of UHMWPE, however, it is clear from Table 1 and Figure 9 that the swelling degree is quite limited. I think the higher porosity of the membranes with longer swelling time at 106°C is caused by the follow-up phase separation of UHMWPE with solvents and crystallization of UHMWPE, as well as the runoff or extraction of amorphous phase in UHMWPE.
8. The data of M1-M5 in Figure 4 is not discussed.
9. As shown in Table 4, why are membranes with higher crystallinity were less thermostable?
10. There are still some typos or grammar mistakes, colloquial expressions, and unreasonable paragraph in this work. Pls double check the texts throughout the manuscript. Moreover, the language should be improved and the logic must be reorganized.
Author Response
Response to Reviewer 3 Comments
Point 1: There is an extensively studied method based on swelling to produce UF membranes, which is termed as "selective swelling-induce pore generation" of block copolymers. See Ref. Acc. Chem. Res., 2016, 49, 1401. This should be included in the Introduction part.
Response 1: Thank you for the information. Indeed, it is correlated with this research, and the paper was adjusted accordingly including these works.
Added text 1: The phenomenon of selective swelling of block-copolymers was implemented by Wang et al. [33–36] and other researchers [37,38] to form a variety of porous structures. In the process of so-called selective swelling-induced pore generation block-copolymer samples are immersed into a swelling agent with a high affinity to the minority block but almost inert to the majority block. The minority block regions expand due to the swelling and then, after evaporation of the solvent (deswelling), the porous structure remains with uniform pore size and straight pore profiles. (In Introduction Section)
In a way, the pore formation mechanism is very similar to the selective swelling induced pore generation in block-copolymers [33–38]. In the case of UHMWPE homopolymer, the crystalline regions of the polymer act as impermeable blocks while the solvent selectively swells the amorphous regions. The difference, however, is that both crystalline and amorphous regions of the SC polymer are different only in terms of their physical state, not their chemical structure. Thus the transformation of one into the other is possible and their ratio changes in both the swelling and deswelling process. In the swelling stage, the crystalline regions are destroyed and transformed into amorphous regions due to temperature and swelling pressure produced by the solvent. Then, in the deswelling stage, some amount of the polymer in the amorphous regions crystallizes as temperature decreases and the solvent is removed. (In Section 3.8)
Point 2: There were no rejection results; these should be added to confirm thus-produced membranes were really UF membranes as the authors claimed.
Response 2: Thank you for the suggestion. We have introduced the data on selectivity of the membranes into the text.
Added text 2: Filtration experiments were carried out in a dead-end stirred filtration cell. For filtration experiments, the membrane coupon was placed onto the porous stainless steel disks and sealed with a rubber O-ring. The active membrane area in the cell was 7.9 cm2. The system was pressurized with helium. The transmembrane pressure was 5 bar. Water was used as a solvent. Blue Dextran (Mw=70 kg/mol) was used as a solute with a concentration of 10 mg/L. Due to the hydrophobic nature of UHMWPE ethanol was filtered through every membrane coupon for 1 hour at 5 bar to fill all membrane pores. Pure water was filtered for 2 hours at 5 bar to determine pure water permeance. Then Blue Dextran solution in water was filtered until 100 ml of permeate was collected. In the case of solution filtration, the feed was stirred at 550 rpm to minimize the concentration polarization effect. The membrane permeance L (L/(m2·h·bar)) was determined as:
|
L=m/(ρ∙S∙t∙∆p), |
(8) |
where ∆p is the trans-membrane pressure (bar), m is the mass of the permeate (g), ρ is the density of the permeate (g/L), S is the active membrane area (m2), and t is the filtration time (h).
The concentration of Blue Dextran in the feed and permeate was measured with a spectrophotometer at the wavelength of 620 nm. The rejection R was calculated using the relation:
|
R=(1-Cp/Cf)·100%, |
(9) |
where Cf and Cp denote the solute concentrations in the feed and permeate respectively. (In Section 2.2.5)
Filtration of Blue Dextran solution demonstrated high rejection for membranes M1 and M2. Analysis of permeate samples after filtration through such membranes showed that dye content was lower than random measurement errors. Further increase in membrane porosity and pore size leads to a rejection decrease down to 22% for membrane M5. (In Section 3.7)
Point 3: The porous regions on the swelling-treated films were present scarcely and randomly. The reviewer thinks the pores may be caused by the dissolution of some low-molecular-weight PE. This possibility should be discussed. Pls compare the weight of the sample before and after swelling. If the weight is reduced after soaking, the dissolution occurs.
Response 3: Thank you for the comment. Although large, well-visible pores on the surface of membranes M1 and M2 are indeed scarce and randomly dispersed (Figure 8), the morphology of cross-section of these samples is quite uniform and contains only very small pores (Figure 9). We agree with the reviewer that large pores in the surfaces of these samples were probably formed due to extraction (dissolution) of some material from the film. As stated in line 487 (please see in the revised unmarked version), these regions are probably located in surface defects of initial film. In such defects swelling (and consequent dissolution) was much faster and thus locally more porous regions were produced. As for the sample weight loss, it was recorded to be 5-20 wt.% depending on the swelling time as stated in lines 370-371. Thus partial dissolution (or extraction of macromolecules that did not contribute to any crystalline region mostly on the lower end of molar mass distribution) occurred in the membrane formation process. It was taken into account for the following swelling degree and porosity calculations.
Point 4: Despite the emphasis of swelling of UHMWPE dense films, this method is in fact based on the thermal induced phase separation (TIPS), but the authors did not clearly distinguish their work with normal TIPS. The authors should explicitly emphasize the superiority of their work than TIPS, otherwise they should not propose such a new method.
Response 4: Thank you for the comment. The mechanism of pore formation is indeed entirely based on thermally induced phase separation (TIPS) while swelling was only a way to prepare membrane precursor (swollen semicrystalline thin film (M1, M2) or homogeneous mixture of UHMWPE with o-xylene in a form of thin film (M3-M5). In the latter case the method is actually standard thermally induced phase separation. Note, however, that preparation of such a membrane precursor is not a trivial task due to very high viscosity of its solutions. In this case swelling of the previously formed dense film might be much easier than trying to prepare homogeneous solution and to form it into desired shape. Therefore the difference between standard TIPS and our approach is that we do not form polymeric solution by trying to dissolve the polymer. Instead we used industrially produced dense film and smaller amount of solvent to swell the film. We emphasized this fact in the paper text (see lines 615-619). Also please note that the fact that TIPS governs the structure formation process was stated several times throughout the manuscript (see, for example, lines 297,538).
Added text 4: In this case, the mechanism of structure formation is not different from the standard TIPS mechanism [18,31,32,40,41]. However, the proposed method could be very useful if for some reason it is difficult either to obtain a homogeneous mixture of the polymer and the solvent or to form it into the desired shape, which is the case for UHMWPE because of the very high viscosity of its solutions. (in Section 3.8)
Point 5: The authors mentioned in line 450 “... probably located in the defects of initial film....”, so it is meaningless to measure the permeance or evaluate the permeance as functions of porosity because the defects are disparate for each membrane.
Response 5: Thank you for the comment. The defects under discussion are located on the surface of the initial film and do not propagate all the way through the bulk of the sample as seen from the SEM images of the cross-sections. Thus, such defects (with sizes up to several tens of microns) do not affect the overall porous structure that defines the transport properties of the membranes. Indeed if these defects were in control of transport properties, it would be impossible to detect through pores with sizes of several tens of nanometers.
Added text 5: The defects, however, do not propagate all the way through the thickness of the sample and thus do not affect the overall transport properties of the membranes.
Point 6: In line 128, the swelling degree (Q) and porosity (P) values were calculated using volumes of initial, swollen, and dried samples, but the authors did not introduce how to test the volumes of samples.
Response 6: The volumes of the samples were calculated by multiplying their dimensions. Since the initial film is industrially produced, its thickness was quite uniform (error less than ±5μm). Thickness of swollen, and dried samples were determined in several spots using a micrometer. It should be noted that all the dimensions (length, width and thickness) increased proportionally within experimental error. We have added this information into the paper text.
Added text 6: The volumes were calculated by multiplying the dimensions of the samples. The thickness of the initial film, and both swollen and dried samples were obtained as the average of at least 10 measurements using a micrometer in different spots. The deviation for the initial film with a thickness of 155μm did not exceed 5 μm, while for the swollen and dried samples it was not greater than 10 μm. Both length and width of the samples were measured using a ruler with an accuracy of 0.5 mm. It should be noted that all the dimensions (length, width and thickness) of the samples increased proportionally during their swelling within an experimental error. (In Section 2.2.2)
Point 7: The authors claimed this work is based on the swelling of UHMWPE, however, it is clear from Table 1 and Figure 9 that the swelling degree is quite limited. I think the higher porosity of the membranes with longer swelling time at 106°C is caused by the follow-up phase separation of UHMWPE with solvents and crystallization of UHMWPE, as well as the runoff or extraction of amorphous phase in UHMWPE.
Response 7: Thank you for the comment. Swelling degrees attained in the membrane formation experiments were in 1.7–11 range. At this, for swelling degrees of 4 and higher according to DSC data the polymer became fully amorphous at 106°C. Nevertheless, whatever were the values of swelling degree, we think that the method of membrane precursor preparation is still based on swelling phenomenon. As stated above in response 4, the mechanism of pore formation, however, is entirely based on TIPS. Thus, we agree that for the membranes obtained from films with longer swelling time the porous structure is a result of polymer crystallization and phase separation of the liquid mixture of UHMWPE and o-xylene. Please note that it was stated in the manuscript (see line 530).
As for the extraction of amorphous “phase”, the following points need to be acknowledged. The sample weight loss was recorded to be 5-20 wt.% depending on the swelling time as stated in lines 342-343. We agree that extraction of amorphous material could result in formation of microregions with lower polymer concentration and thus lead to locally higher porosity. However, this process must not be thought as similar to pore formation by extraction of soluble component from the insoluble matrix. In semicrystalline polymers (especially with such a high molecular weight) the same macromolecules contribute both to the amorphous (tie chains, loops, cilia) and crystalline (strands within the lamellae) regions. Thus as one block of the multiblock-copolymer cannot be extracted from the sample, not all the amorphous material can be extracted from the semicrystalline polymer sample. Instead only the macromolecules that do not contribute any crystallite can be removed from the sample by extraction. The difference from the block-copolymer sample is that crystalline regions become amorphous (melt) due to increased temperature and presence of the solvent. Therefore, some macromolecules that initially took part in one or several crystallites become free and obtain ability to be extracted. Taking the aforementioned into the account, it can be concluded that the process of extraction of amorphous material is not a defining factor in pore formation process.
Point 8: The data of M1-M5 in Figure 4 is not discussed.
Response 8: This data was discussed in section 3.3. However, the discussion of this this data was too far from the place where the Figure 4 first appears. Thus taking point 10 into account we reorganized logic and placed Figure 4 closer to its main point of discussion.
Point 9: As shown in Table 4, why are membranes with higher crystallinity were less thermostable?
Response 9: Thank you for the interesting question. In the present work, the thermostability was assessed by measuring the shrinkage of the membranes after their annealing for 1 h at different temperatures. From the obtained data it follows that the samples with higher crystallinity degree (more porous ones) tend to shrink more and at lower temperatures. However, during an analysis of these data, it must be accounted that all the membranes underwent some shrinkage already at the cooling, extraction, and drying stages, as stated in lines 385-394. Simple calculations using the data from Table 2 show that the volume shrinkage of these samples during these stages of the membrane formation process was ~104, 76, 53, 44, and 37% for samples M1, M2, M3, M4, and M5, respectively. In other words, the shrinkage (which is due to the relaxation of the internal stresses in the sample) during the membrane formation stages decreases as the crystallinity degree (and porosity) of the membranes increases. This fact agrees well with data of [see, for example, 10.1002/mats.201300147] according to which the relaxation times increase with the increase of the polymer degree of crystallinity. Bearing this in mind it can be assumed that during the time of extraction and drying of the samples the relaxation proceeded more deeply in the samples with a lower degree of crystallinity. That is probably the reason why the samples with higher crystallinity degree tend to shrink more and at lower temperature than the samples with lower crystallinity degree. The decrease of crystallinity degree during heating (in experiments on thermostability) of these samples opened the otherwise frozen possibility for their relaxation and thus these samples started to shrink.
Point 10: There are still some typos or grammar mistakes, colloquial expressions, and unreasonable paragraph in this work. Pls double check the texts throughout the manuscript. Moreover, the language should be improved and the logic must be reorganized.
Response 10: We improved our English throughout the paper and reorganized some fragments of the manuscript.
Round 2
Reviewer 3 Report
The revision can be accepted. But the spelling of the authors name was wrong in Ref 35.